# The Effectiveness of Cigarette Pack Health Warning Labels with Religious Messages in an Urban Setting in Indonesia: A Cross-Sectional Study

**DOI:** 10.3390/ijerph16214287

**Published:** 2019-11-05

**Authors:** Bekir Kaplan, Jeffrey J. Hardesty, Santi Martini, Hario Megatsari, Ryan D. Kennedy, Joanna E. Cohen

**Affiliations:** 1Department of Health, Behavior and Society, Institute for Global Tobacco Control, Johns Hopkins Bloomberg School of Public Health, 615 N Wolfe St, Baltimore, MD 21205, USA; 2Faculty of Public Health, University of Airlangga, Jl. Mulyorejo, Surabaya 60115, Indonesia

**Keywords:** spiritual messages, tobacco control, Indonesia

## Abstract

This study sought to assess the effectiveness of religious cigarette health warning labels (HWLs) in Indonesia, a country with a high public health burden from tobacco use. The study tested different religious and nonreligious messages related to suicide, secondhand smoke (SHS) and gangrene. Participants were smokers and non-smokers from Surabaya, Indonesia (*n* = 817). Participants rated each HWL for its effectiveness on a scale of 1 to 10 (1 = “not at all”, 10 = “extremely”) with respect to 10 items. Nonreligious HWLs were marginally superior for SHS and suicide while religious HWLs were marginally superior for gangrene. Given the close rating scores between religious and nonreligious HWLs, they were functionally equal in effectiveness. With proper assessment of potential unintended consequences, the implementation of religious HWLs could be considered for a proportion of HWLs.

## 1. Introduction

Indonesia has over 60 million smokers, the third-highest number of smokers among all countries in the world [1]. Over the last two decades, smoking prevalence in Indonesia has increased nearly 25% (from 54% to 67%) among males and 260% (from 1.7% to 4.5%) among females [1]. Compared to other countries in the region, Indonesia has minimal tobacco control measures at the national level and is one of the few countries in the world that has not signed or ratified the World Health Organization (WHO) Framework Convention on Tobacco Control (FCTC). The WHO FCTC sets minimum standards for impactful tobacco control policies, including health warning labels (HWLs) on cigarette packs. Despite a lack of participation in the global treaty, Indonesia has made progress in tobacco control, including developing and implementing one set of five cigarette pictorial health warning labels introduced in 2014 [2].

Indonesia has the largest population of Muslims in the world, with 86% of Indonesians declaring themselves as Muslims [3]. The Indonesia Family Life Survey-4 reported that 79% of Indonesians are highly religious [4]. Moreover, religion had a particular effect on the recent election results, suggesting that religion plays a significant role in the political, economic and cultural spheres in Indonesia [5,6,7].

The relationship between religion and public health has been a longstanding interest in the health, social, and behavioral sciences [8]. Religion can affect health status by establishing norms regarding behavior, including health-related behaviors [9].

There is evidence in the public health literature that religion can play a substantial role in health beliefs and behaviors [10,11,12,13,14]. Several studies reported that higher religiosity level was associated with lower tobacco, alcohol, and illicit drug use [10,11,12,13,14]. Regarding tobacco control, Ramadan (a holy month) could be used to support people to quit smoking [14]. Furthermore, the WHO called upon its Member States to include a spiritual dimension in their health strategies [15] and encourages engaging with religious leaders to advance tobacco control priorities [16].

Reducing tobacco use in Indonesia will require a change in social norms. Given the substantial role of religion in Indonesia, cigarette health warning labels (HWLs) with religious messages may help to de-normalize tobacco use in Indonesia [1,4]. In this study, we aimed to assess the effectiveness of religious HWLs among Muslims in Surabaya, Indonesia, a city with a population of 3.5 million.

## 2. Methods

### 2.1. Sample and Recruitment

In this cross-sectional study, two teams supported data collection from 28 April to 2 May 2018, in two malls in Surabaya, the second-most populous city on Java Island, Indonesia. Potential participants were recruited via intercept then screened for eligibility based on age (≥18 years), religion (Muslims), and length of residency (≥1 year) in Surabaya. Quotas were utilized for smoking status (80% smokers, 20% nonsmokers) and sex (80% male, 20% female) to oversample female smokers. Once completed, survey participants received a voucher for 100,000 IDR (~$6.25 USD) for their time.

### 2.2. Health Warning Label Design

To create religious-based HWLs, we used a WHO publication [17] regarding the Islamic view on smoking as a guide. Islam works to reduce harm to society and individuals. Many principles of Islam call upon people to look after their own and others’ health, to avoid health hazards and risks [17]. For instance, suicide, harming yourself, and harming others are forbidden in Islam and these behaviors could be relevant to smoking given the harmful effects of cigarettes on health [17]. Considering the Islamic view on smoking, we developed three messages as follows: (1) Smoking kills you gradually (Gradual Suicide) and suicide is haram (forbidden), (2) Smoking around others disturbs and violates other people. Disturbing other people is dzolim (cruelty), and, (3) Smoking causes gangrene and damaging yourself is haram (forbidden) (Figure 1). The featured text and image pairings are designed to elicit feelings of guilt from self-harm (suicide and gangrene) and harming others (secondhand smoke). We obtained necessary permissions to use the death and SHS images from the Centers for Disease Control and Prevention and the gangrene image from Commonwealth of Australia.

To compare the effectiveness of religious messages, we developed three corresponding nonreligious messages as follows: (1) Smoking harms your health and can kill you, (2) Smoking harms other people and causes fatal lung diseases in nonsmokers, and, (3) Smoking causes gangrene and harms blood vessels and cause gangrene (Figure 2). 

### 2.3. Rating the Effectiveness of HWLs

The six HWLs were shown to participants in random order. All participants rated each HWL on a scale of 1 to 10 (1 = “not at all”, 10 = “extremely”) for whether the HWL: (1) grabs my attention, (2) is easy to understand, (3) is credible, (4) makes me feel more concerned about smoking, (5) makes me think about the health risks of smoking, (6) motivates me to stay smoke-free (for nonsmokers), (7) makes me think about quitting (for smokers), (8) motivates me to quit smoking (for smokers), (9) makes me avoid looking at the warning label, and, (10) the effectiveness of the HWL.

We also created an “average overall rating score” for religious HWLs by summing the values of the 10 rating scores for each religious HWL, and dividing by the total number of rating questions (30), the same procedure was used for the nonreligious HWLs. The Cronbach’s alpha for the six HWL questions was 0.961.

### 2.4. Religiosity Level

Religious predisposition was measured using the Centrality of Religiosity Scale (CRS) [18]. The validation study of CRS was conducted in Indonesia by Wardhani and Dewi [19]. The CSR measures five theoretical core dimensions of religiosity, including public practice, private practice, religious experience, ideology and intellectual (thoughts). The CRS assessed each dimension using three levels (1 or 2 = not at all, 3 = moderately, 4 or 5 = quite a bit/very much so).

### 2.5. Smoking Behavior and Intentions

Smoking status was measured using questions from the Global Adult Tobacco Survey-Indonesia [1]. Participants were defined as “daily smokers” if they smoked every day, “occasional smokers” if they smoked less than daily but more than 100 cigarettes in their entire life. Never smokers, former smokers, and smokers smoked fewer than 100 cigarettes in their entire life were defined as non-smokers. Intention to quit was assessed by asking ‘‘Are you planning to quit in the next month, 6 months, beyond 6 months, or not at all?” Past year quit attempts was coded as a dichotomous variable reflecting at least one attempt to quit in the last year (1) or none (0). Time for the first cigarette in the morning was measured using the following choices: “within 5 min,” “6–30 min,” “31–60 min,” and “more than 60 min [20].” Motivation to quit for smokers and motivation to stay smoke-free for nonsmokers were rated by the participants on a scale of 1 to 10 (1 = “not at all”, 10 = “extremely”). All sociodemographic and smoking behavior questions were asked before the rating questions.

### 2.6. Statistical Analysis

Statistical analyses were performed using SPSS^®^ for Windows^®^, version 21.0 (SPSS Inc., Chicago, IL, USA). Differences between rating scores were tested with the Wilcoxon signed-rank test. Spearman correlation and linear regression analyses were conducted to assess potential associations between participants’ religiosity level and their HWL rating scores. The beta coefficients and 95% confidence intervals from each linear regression model were exponentiated to obtain geometric mean ratios (GMRs) and corresponding 95% confidence intervals (CI). The HWL rating scores and the religious and nonreligious overall rating scores were left-skewed and log-transformed for the linear regression analyses. All statistical tests were two-sided (α = 0.05).

Institutional Review Boards (IRB) at the Johns Hopkins Bloomberg School of Public Health in Baltimore (United States of America) (IRB No: 8315) and at the University of Airlangga Faculty of Public Health in Surabaya (Indonesia) approved the study protocol.

### 2.7. Ethics Approval

The study was approved by the Institutional Review Boards of the Johns Hopkins Bloomberg School of Public Health and the Institutional Review Boards of University of Airlangga, Faculty of Public Health.

## 3. Results

In total, we screened 1045 potential participants, 11 (1.1%) resided in the survey city less than one year, 26 (2.5%) were non-Muslim, 11 (1.1%) were less than 18 years old, and 180 (17.2%) did not meet the quotas. Our final sample was comprised of 817 participants.

The study population of 817 participants were 78.8% male, 51.5% were 18–24 years old, 62.7% had graduated high school, 33.6% had children, 29.9% reported smoking their first cigarette within 5 min of waking, 54.8% reported their motivation to quit was at least a 6 on the 10-point scale, and, 54.6% and 29.7% reported good and excellent health status, respectively. Among smokers, 51.0% attempted to quit in the last year and 36.3% did not plan to quit smoking in the future. Most participants (86.2%) had a high CSR score (4 or 5), 12.0% a moderate score (3), and 1.8% a low score (1 or 2) (Table 1).

### 3.1. Sociodemographic Characteristics of Participants by Religiosity Level

College graduates and occasionally/not at all smokers were more religious than primary/secondary graduates and daily smokers (*p* = 0.049 and *p* = 0.004, respectively). Participants who reported very good /excellent health status (88.9%) and between 3–5 million IDR monthly income (91.2%) had higher religiosity scores than participants who reported poor/fair health status (76.4%) and less than 3 million IDR monthly income (84.0%) (*p* = 0.001 and *p* = 0.040, respectively). There were no significant differences between religiosity level and gender, age, and having children (Table 2). In addition, we did not find any significant difference between the rating score of religious and nonreligious HWLs and the sociodemographic characteristics of the participants.

### 3.2. Rating Scores of Religious and Nonreligious HWLs

The nonreligious HWLs had a median rating score of 7.3, which was slightly higher than the religious HWLs’ median score of 7.2. (*p* = 0.005) (Table 3).

The mean values of the nonreligious suicide HWL, with regards to credibility and general effectiveness (7.4 ± 3.0 and 7.1 ± 3.1, respectively), were higher for than the religious suicide HWL (7.1 ± 3.1 and 6.9 ± 3.2, respectively) (*p* = 0.001 and *p* = 0.016) (Table 3). There were no statistically significant differences between the nonreligious and religious suicide HWLs with regards to following rating questions: grabs my attention, easy to understand, makes me feel more concerned about smoking, makes me think about health risk of smoking, motivates me to stay smoke-free (for nonsmokers), makes me think about quitting (for smokers only), motivates me to quit smoking (for smokers only), and makes me avoid looking at the warning label (Table 3).

The nonreligious secondhand smoke (SHS) HWL was rated significantly higher than the religious SHS HWL with regard to credibility, makes me feel more concerned about smoking, makes me think about the health risks of smoking, makes me think about quitting (for smokers only), motivates me to quit smoking (for smokers only), makes me avoid looking at the warning label, and the effectiveness of the HWL (*p* < 0.05). There were no statistically significant differences between the nonreligious and religious SHS HWLs with regard to grabs my attention, easy to understand, and motivates me to stay smoke-free (*p* > 0.05) (Table 3).

The mean value of the religious gangrene HWL (7.0 ± 3.2) was higher than the nonreligious gangrene HWL (6.9 ± 3.3) (*p* = 0.042) with regard to making me think about the health risks of smoking. There were no significant differences between the religious and nonreligious gangrene HWLs for the other rating questions (Table 3).

In response to the credibility outcome, the nonreligious SHS HWL had the highest mean rating score (8.00), followed by the religious SHS HWL (7.7) and the nonreligious suicide HWL (7.4). The mean values of the SHS HWLs were higher than suicide and gangrene HWLs with regard to credibility (Table 3).

### 3.3. The Association between CRS Score and HWL Rating Scores

There were significant but weak (ρ: 0.102–0.187) correlations between the CRS score of the participants and nine of the 10 rating questions. The only rating question not correlated with CRS was “motivates me to stay smoke-free” which had a median score “10” for all HWLs among nonsmokers. In addition, the participants with high CRS scores (4 or 5) rated all ten rating questions higher than the participants with low/moderate CRS scores (1 to 3) (*p* < 0.05).

There were significant associations between the credibility of the HWLs and the CRS scores in the linear regression models. Each additional unit increase in the CRS score was associated with a 5 to 8 percent increase in the credibility of HWLs after adjustment for age, gender, income, education status, health status, having children, how carefully the participant reported s/he read the HWL, motivation to quit for smokers, and motivation to stay smoke free (Table 4).

Each additional unit increase in the CRS was associated with a 4% increase in the rating score of the religious suicide and SHS HWLs with regard to making them more concerned about smoking, and a 5% increase in the rating score of the same HWLs with regard to making them think about quitting (Table 4).

Each additional unit increase in the CRS score was associated with a 4% increase in the rating score of the religious suicide HWLs with regard to motivating people to quit smoking (Table 4).

## 4. Discussion

This is the first study to examine the effectiveness of religious messages on pictorial HWLs. The nonreligious HWLs were marginally superior with the SHS and suicide themes, while the religious HWL was marginally superior with the gangrene theme. The religious and nonreligious SHS HWLs were more credible than the suicide and gangrene HWLs. The religious gangrene and suicide HWLs were as effective as the nonreligious HWLs with regards to grabbing attention, easy to understand, making people more concerned about smoking, motivating people to stay smoke-free, making people think about quitting, and motivating people to quit smoking. Given the close rating scores between the religious and nonreligious HWLs, religious and nonreligious HWLs were functionally equal in effectiveness. With proper assessment of the potential unintended consequences, implementation of religious HWLs could be considered for a proportion of HWLs in countries where religion has a substantial effect on social life.

To date, several studies have shown the potential benefits of religious-based tobacco control interventions [21,22,23]. In a national survey in the US [21], it was reported that regular religious attendance might serve as a protective factor against cigarette smoking among black men. In Malaysia, it was shown that religious norms may play a greater role than secular norms in influencing quit attempts among Muslim Malaysian and Buddhist Thai adult smokers [22]. Another study in Aceh-Indonesia reported that school-based smoking prevention programs could be more effective if they include Islamic teachings and rulings on tobacco smoking [23]. Other studies have shown that religiosity reduces the risk of tobacco smoking and other risky behaviors and that religiosity/spirituality can have positive effects on behaviors relating to health [10,11,12,13,14].

The use of religion to improve health is not new. Previous research indicates that religion can facilitate successful quitting among adults. In a cross-sectional study in Saudi Arabia, religious considerations were the most important reasons for not smoking among never-smokers, for quitting among ex-smokers, and for attempting to quit or thinking about quitting among current smokers [24]. In an experimental study among Buddhists in Thailand, trying to quit smoking and stopping smoking for more than 1 year were greater among religious people compared to less religious [25]. Previously, religious messages have been used to help reducing smoking in several Muslim countries such as Egypt and Saudi Arabia [26]. An awareness campaign in Saudi Arabia including religious messages about smoking led to 200 retailers no longer selling tobacco [26]. Given the themes used in the religious messages in this study (suicide, harming yourself and others) are discouraged by many religions, similar messages could be tested for other faiths, where deemed appropriate, to determine if they help increase knowledge about the harms of smoking and encourage smoking cessation.

The central aims of the Islamic legal framework, as well as a majority of other religions, are to minimize the risk of harm to society and individuals and maximize the opportunities for collective and individual wellbeing. Considering this framework, we created three different text messages consisting of suicide, harming yourself, and harming others. All of these are prohibited behaviors in Islam and could result from smoking. The created religious HWLs are designed to elicit feelings of guilt from self-harm (suicide and gangrene) and harming others (secondhand smoke).

All HWLs in the current study were more effective among those with high religiosity scores. To the best of our knowledge, this is the first study to report that cigarette HWLs are more effective among religious Muslims in Indonesia compared to Muslims with low or moderate) religiosity scores.

Religious-based tobacco control interventions have been targeted by tobacco companies. It was reported in tobacco company internal documents that religion-based tobacco control activities are a significant threat to their expansion into emerging markets [27]. In order to counter religious-based tobacco control interventions, tobacco companies framed Islamic objection to smoking as “extremism [27].” In 1985, a lobbyist advising a tobacco company stated, “*A Moslem who attacks smoking generally speaking would be a threat to existing government as a “fundamentalist” who wishes to return to Sharia law…*” thereby indicating tobacco companies’ strategy with religious objection to smoking [28].

The credibilities of the SHS HWLs were higher than the suicide or gangrene HWLs. One potential reason could be a 2009 fatwa (religious rulings or opinion) issued by Majelis Ulama Indonesia (MUI), the government-funded council in Jakarta that includes representation from many Indonesian Muslim organizations [16]. The fatwa announced “*smoking in public and smoking [near] children or pregnant women is haram (prohibited)*”, otherwise smoking was said to be makruh (discouraged, not sinful but those abstaining from it will be blessed by God) [16]. In this announcement, there is a clear denouncement of SHS exposure from smoking, which might have affected the participants’ responses to the SHS HWLs in the current study. Furthermore, recent news reporting the harmful effect of SHS on children in Indonesia might have also affected participants’ responses to the SHS HWLs in the current study [29,30]. Another reason could be that the majority of participants in the study were under 25 years of age, these young participants might have perceived illness or suicide as being of less relevance compared to older participants.

Using religious-based HWLs in countries where smoking is perceived as a norm of social life may aid in the denormalization of smoking and could be considered by health authorities as a promising avenue to include in tobacco control efforts. The implementation of religious HWLs could be considered for a proportion of HWLs. First, it would be important to assess the magnitude of potential unintended consequences. For instance, there is a need to study and consider whether religious HWLs could have negative impacts on people of other faiths or people of no faith. Furthermore, additional evidence is needed to determine whether religious HWLs cause people, particularly older unhealthy adults and those trying but failing to quit, to associate their poor health outcome with non-adherence to religious practices. Given the close ties between religion and personal identity, inducing significant feelings of guilt related to non-adherence could have positive or negative implications for smoking and/or mental health. Finally, advocating for religious HWLs will likely involve partnering with religious institutions and authorities, whose public arguments tend to be religious and political in nature rather than health-focused. Given their ties to specific political parties or factions of politicians, partnering could be beneficial in the near-term, however there may be long-term risks. By amplifying the voice of religious institutions and authorities, public health institutions could be promoting their social standing as an authority on health-related policy, which may cause challenges in the future when interests do not align.

This study has some potential limitations. First, the HWLs were not tested on people of faiths other than Islam. That being said, suicide, secondhand smoke, and harming yourself are generally important themes for many religions. Similar HWLs could be created and tested with people of other faiths. Therefore, we can only make conclusions about our sample of Muslims. Second, we tested the textual religious messages with images that were already in use in other countries. Thus, we could have better-matched images with the religious textual messages if we created more specific graphical images in accordance with the text. Third, we used sex (80% male, 20% female) and smoking status (80% smokers, 20% non-smokers) quotas to obtain a more representative sample of the individuals that would be affected by HWLs, but did not have age quotas, resulting in a large number of younger participants and an underrepresentation of people aged 45 and older. Fourth, participants were recruited in an urban setting in Indonesia. Despite the high religiosity level, the participants in the current study might have more secular norms than those who live in rural settings in Indonesia. Fifth, in order to measure the effectiveness of the health warning labels, participants had a one time, forced exposure to health warning labels, which differs substantially from prolonged, naturalistic exposure. Despite this survey method being used previously in numerous studies [25,31,32] to measure the effectiveness of health warning labels, other study methods that provide more time to participants to review health warning labels could be better for measuring real-world effectiveness. Sixth, we did not test the effectiveness of textual messages and graphical images separately. Therefore, the current study was not able to differentiate the observed effect from the graphic pictures themselves and the addition of religious textual messages. Seventh, this was an exploratory study tested under specific conditions for a specific group of people. Before a recommendation can be made about using religious messages on HWLs, it would be important to assess the magnitude of potential unintended consequences. For instance, there is a need to study and consider whether religious HWLs could have negative impacts on people of other faiths or people of no faith, particularly in states where power is concentrated amongst a single demographic or ruling class (e.g., persons of a single religion).

## 5. Conclusions

The nonreligious HWLs were marginally superior with the SHS and suicide themes while the religious HWL was marginally superior with the gangrene theme. Given the close rating scores between the religious and nonreligious HWLs, the religious and nonreligious HWLs were functionally equal in effectiveness. The SHS HWLs were more credible than the suicide and gangrene HWLs. With additional consideration of possible unintended consequences, religious-based HWLs could be considered for use in countries where religious norms have a large role in social life. Considering other harmful health effects of smoking, this work could be expanded by additional religious textual messages stating other health risks of tobacco in combination with appropriate graphical images and tested with regard to their effectiveness.

## Figures and Tables

**Figure 1 ijerph-16-04287-f001:**
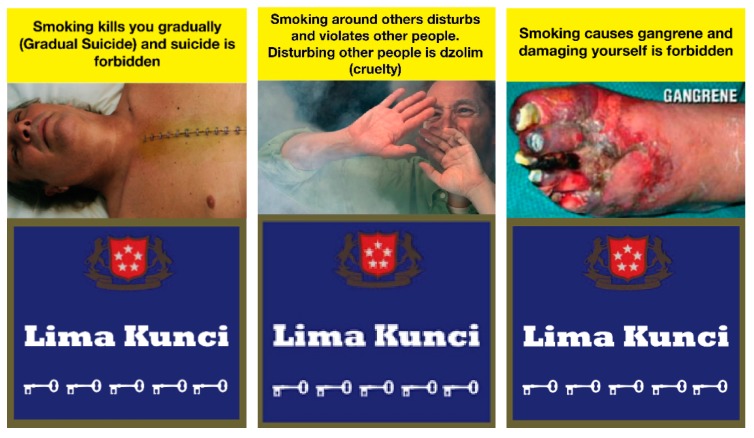
Religious Health Warning Labels.

**Figure 2 ijerph-16-04287-f002:**
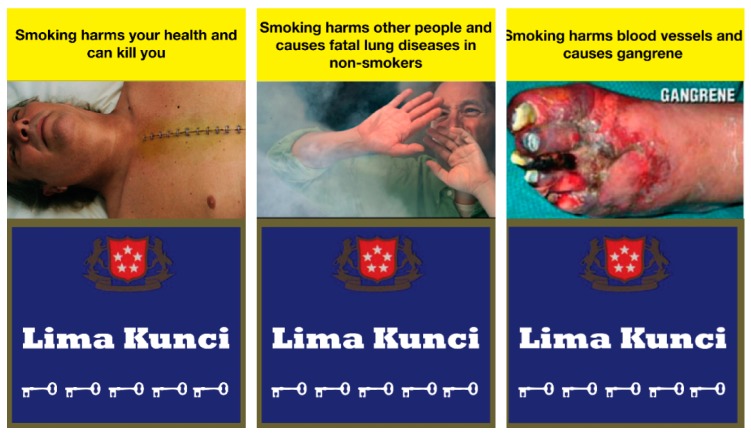
Nonreligious Health Warning Labels.

**Table 1 ijerph-16-04287-t001:** Sociodemographic Characteristics of Participants.

**Gender**	***n***	**%**
Male	644	78.8
Female	173	21.2
**Age (Year)**		
18–24	421	51.5
25–34	236	28.9
35–44	103	12.6
≥45	50	6.1
**Education Level**		
Primary/Secondary	76	9.3
High School	510	62.7
College	228	28.0
**Having Children**		
No	538	66.4
Yes	272	33.6
**Monthly Income (IDR)**		
≤3 million IDR	420	54.9
>3 million and ≤5 million IDR	217	28.4
>5 million IDR	128	16.7
**Smoking Status**		
Daily	642	78.6
Occasionally	46	5.6
Not at all	129	15.8
**Health Status**		
Poor/Fair	123	15.6
Good	430	54.6
Very Good/Excellent	234	29.7
**Religiosity Level (*n* = 665)**		
Quite a bit/very much so (4 or 5)	706	86.6
Moderately (3)	93	11.4
Not at all (1 or 2)	16	2.0
**Daily Practice**		
5 or more	497	60.8
1–4 times in a day	280	34.3
Never	40	4.9
**Time to First Cigarette (*n* = 633)**		
Within 5 min	189	29.9
6–30 min	253	40.0
More than 30 min	191	30.2
**Quit Attempt in the Past Year (*n* = 635)**		
Yes	324	51.0
No	311	49.0
**Plan to Quit (*n* = 634)**		
Within next month	119	18.8
Within next 6 months	136	21.5
Beyond next 6 months	149	23.5
Not at all	230	36.3
**Motivation to Quit (*n* = 635) ***		
1–5	287	45.2
6–10	348	54.8
**Motivation to Stay Smoke Free (*n* = 172) ***		
1–7	80	46.5
8–10	92	53.5
**Total**	817	100.0

* The median values were used as cut off points.

**Table 2 ijerph-16-04287-t002:** Sociodemographic Characteristics of Participants by Religiosity Level.

Characteristics	Religiosity Level				*p*
	Score 1–3		Score 4–5		
	*n*	%	*n*	%	
**Gender**					
Male	84	13.0	560	87.0	0.590
Female	25	14.6	146	85.4	
**Age (Year)**					
18–24	65	15.5	354	84.5	0.065
≥25	44	11.1	352	88.9	
**Education Level**					
Primary/Secondary	16	21.1	60	78.9	0.049
High School	69	13.6	440	86.4	
College	23	10.1	205	89.9	
**Having Children**					
No	74	13.8	462	86.2	0.606
Yes	34	12.5	238	87.5	
**Monthly Income (IDR)**					
≤3 million	67	16.0	352	84.0	0.040
>3 million and ≤5 million	19	8.8	198	91.2	
>5 million	17	13.3	111	86.7	
**Smoking Status**					
Daily	97	15.2	543	84.8	0.004
Occasionally/Not at all	12	6.9	163	93.1	
**Health Status**					
Poor/Fair	29	23.6	94	76.4	0.001
Good	48	11.2	380	88.8	
Very Good/Excellent	26	11.1	208	88.9	
**Total**	109	13.4	706	86.6	

**Table 3 ijerph-16-04287-t003:** Average Scores of the HWLs by the Rating Questions.

Rating Questions	*n*	Nonreligious Death		Religious Death			Nonreligious SHS		Religious SHS		Nonreligious Gangrene			Religious Gangrene	
		Mean ± SD	Med	Mean ± SD		Med	Mean ± SD	Med	Mean ± SD	Med	Mean ± SD		Med	Mean ± SD	Med
**Grabs My Attention**	810	6.9 ± 3.3	8.0	6.8 ± 3.3		8.0	7.3 ± 3.0	8.0	7.2 ± 3.1	8.0	7.0 ± 3.3		8.0	7.0 ± 3.2	8.0
	***p*** *****	0.080					0.492				0.528				
**Easy to Understand**	810	7.4 ± 3.0	8.0	7.4 ± 3.0		8.0	7.8 ± 2.8	9.0	7.6 ± 2.8	9.0	7.1 ± 3.1		8.0	7.1 ± 3.1	8.0
	***p*** *****	0.577					0.131				0.489				
**Credible**	804	7.4 ± 3.0	9.0	7.1 ± 3.1		8.0	8.0 ± 2.7	9.0	7.7 ± 2.8	9.0	7.0 ± 3.1		8.0	7.0 ± 3.2	8.0
	***p*** *****	0.001					<0.001				0.704				
**Makes Me Feel More Concerned about Smoking**	807	6.7 ± 3.4	8.0	6.7 ± 3.3		8.0	6.8 ± 3.2	8.0	6.7 ± 3.3	8.0	6.7 ± 3.3		7.0	6.8 ± 3.4	8.0
	***p*** *****	0.436					0.011				0.086				
**Makes Me Think about the Health Risks of Smoking**	804	6.9 ± 3.3	8.0	6.8 ± 3.2		8.0	7.0 ± 3.2	8.0	6.8 ± 3.2	8.0	6.9 ± 3.3		8.0	7.0 ± 3.2	8.0
	***p*** *****	0.141					0.002				0.042				
**Motivates Me to Stay Smoke-Free (Non-Smokers)**	175	8.5 ± 2.5	10.0	8.6 ± 2.4		10.0	8.6 ± 2.4	10.0	8.6 ± 2.4	10.0	8.7 ± 2.3		10.0	8.7 ± 2.3	10.0
	***p*** *****	0.229					0.805				0.911				
**Makes Me Think about Quitting (Smokers)**	628	6.1 ± 3.4	6.0	5.9 ± 3.4		6.0	6.1 ± 3.4	6.0	5.9 ± 3.4	6.0	6.1 ± 3.4		7.0	6.1 ± 3.4	7.0
	***p*** *****	0.076					0.027				0.431				
**Motivates Me to Quit Smoking (Smokers)**	624	6.1 ± 3.5	6.0	6.0 ± 3.4		6.0	6.1 ± 3.4	6.0	5.9 ± 3.4	6.0	6.1 ± 3.4		7.0	6.1 ± 3.4	7.0
	***p*** *****	0.219					0.001				0.490				
**Makes Me Avoid Looking at the Warning Label**	799	5.9 ± 3.5	6.0	5.9 ± 3.5		6.0	5.8 ± 3.5	6.0	5.7 ± 3.5	6.0	6.1 ± 3.5		7.0	6.2 ± 3.5	7.0
	***p*** *****	0.842					0.040				0.985				
**In General, How Effective is This Warning**	797	7.1 ± 3.1	8.0	6.9 ± 3.2		8.0	7.2 ± 3.0	8.0	7.1 ± 3.0	8.0	7.0 ± 3.2		8.0	7.1 ± 3.1	8.0
	***p*** *****	0.016					0.007				0.714				
****Overall Rating Score****		**Average Rating Score of All RELIGIOUS HWLs**							**Average Rating Score of All NONRELIGIOUS HWLs**						
		**Mean ± SD**			**Median**				**Mean ± SD**			**Median**			
	815	6.8 ± 2.6			7.2				6.9 ± 2.6			7.3			
	***p*** *****	0.005													

* Wilcoxon Test to assess whether there was a difference in the median scores.

**Table 4 ijerph-16-04287-t004:** Ratio of Geometric Means of Rating Questions by Centrality of Religiosity Score **.

Rating Questions	*n*	Nonreligious Death	Religious Death	Nonreligious SHS	Religious SHS	Nonreligious Gangrene	Religious Gangrene
		GMR * (95%CI)	GMR * (95%CI)	GMR* (95%CI)	GMR* (95%CI)	GMR* (95%CI)	GMR* (95%CI)
**Grabs My Attention**	810	1.01 (0.98–1.04)	1.04 (1.01–1.07)	1.04 (1.01–1.06)	1.03 (0.99–1.05)	1.01 (0.97–1.03)	1.02 (0.99–1.05)
**Easy to Understand**	810	1.02 (0.99–1.05)	0.99 (0.96–1.02)	0.99 (0.97–1.02)	0.97 (0.95–0.99)	0.99 (0.97–1.02)	1.03 (0.99–1.06)
**Credible**	804	1.05 (1.01–1.09)	1.07 (1.02–1.12)	1.05 (1.02–1.09)	1.07 (1.03–1.11)	1.06 (1.02–1.11)	1.08 (1.04–1.13)
**Makes Me Feel More Concerned about Smoking**	807	1.02 (0.99–1.06)	1.04 (1.01–1.07)	1.04 (0.99–1.07)	1.04 (1.01–1.08)	1.02 (0.99–1.05)	1.02 (0.99–1.05)
**Makes Me Think about the Health Risks of Smoking**	804	1.03 (0.99–1.06)	1.04 (1.01–1.07)	1.05 (1.02–1.09)	1.03 (0.99–1.07)	1.02 (0.98–1.05)	1.02 (0.98–1.05)
**Motivates Me to Stay Smoke-Free (Non-Smokers)**	175	0.98 (0.91–1.05)	1.02 (0.96–1.09)	1.03 (0.96–1.11)	0.97 (0.91–1.04)	1.04 (0.97–1.12)	0.99 (0.93–1.05)
**Makes Me Think about Quitting (Smokers)**	628	1.04 (0.99–1.08)	1.05 (1.01–1.09)	1.03 (0.99–1.07)	1.05 (1.02–1.11)	1.03 (0.99–1.07)	1.02 (0.98–1.06)
**Motivates Me to Quit Smoking (Smokers)**	624	1.03 (0.99–1.08)	1.04 (1.01–1.08)	1.03 (0.99–1.07)	1.04 (0.99–1.09)	1.03 (0.98–1.07)	1.01 (0.97–1.04)
**Makes Me Avoid Looking at the Warning Label**	799	1.02 (0.98–1.07)	1.01 (0.96–1.05)	1.02 (0.97–1.07)	1.01 (0.95–1.05)	1.01 (0.96–1.05)	0.99 (0.95–1.04)
**In General, How Effective is This Warning**	797	1.01 (0.98–1.03)	1.02 (0.99–1.05)	1.02 (0.99–1.05)	0.99 (0.96–1.02)	1.01 (0.97–1.03)	0.99 (0.96–1.02)

* GMRs were adjusted by age, gender, income, education status, health status, having children, credibility, how carefully read the HWL, motivation to quit for smokers, motivation to stay smoke free for nonsmoker. ** The CRS on a scale of 1 to 5.

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
