# Peer review of "The Effectiveness of Cigarette Pack Health Warning Labels with Religious Messages in an Urban Setting in Indonesia: A Cross-Sectional Study"

_ijerph, 2019, doi:10.3390/ijerph16214287_

Round 1

Reviewer 1 Report

Overall I found the study described very interesting.  The use of religion-based messages for health warnings is novel and potentially very impactful.  Two points for consideration are that the results are hardly different between the religion-based labels and the non-religion-based labels.  Therefore, I think more discussion should be included regarding the conclusion that religion-based labels are recommended, particularly given that other religions were not included in the label designs or testing.  Additionally, more discussion would be appreciated regarding the choices of 3 specific messages used and the possible expansion of this work with additional health risks and differences between the specific messages (where one might be more impactful than another).

Author Response

Overall I found the study described very interesting.  The use of religion-based messages for health warnings is novel and potentially very impactful.  Two points for consideration are that the results are hardly different between the religion-based labels and the non-religion-based labels.  Therefore, I think more discussion should be included regarding the conclusion that religion-based labels are recommended, particularly given that other religions were not included in the label designs or testing.

Response: Thank you for this feedback. We added additional paragraph to the discussion regarding the conclusion that religion-based labels are recommended. In addition, it should be noted that it is not our intention to necessarily recommend religious-based labels at this time; however, our data indicate they could be considered as a possible option. As alluded to in our discussion, there are several potential unintended consequences that should be evaluated before a fulsome policy recommendation could be made. We discuss that religious health education messages have been used in several Muslim countries. In addition, we added one more example from Thailand that religiousness among Buddhists has a positive effect on trying to quit smoking as well as stopping smoking for more than one year.

We added the following paragraph to the discussion:

The use of religion to improve health is not new. Previous research indicates that religion can facilitate successful quitting among adults. In a cross-sectional study in Saudi Arabia, religious considerations were the most important reasons for not smoking among never-smokers, for quitting among ex-smokers, and for attempting to quit or thinking about quitting among current smokers.1 In an experimental study among Buddhists in Thailand, trying to quit smoking and stopping smoking for more than 1 year were greater among religious people compared to less religious.29  Previously, religious messages have been used to help reducing smoking in several Muslim countries such as Egypt and Saudi Arabia.2 An awareness campaign in Saudi Arabia including religious messages about smoking led to 200 retailers no longer selling tobacco.2 Given the themes used in the religious messages in this study (suicide, harming yourself and others) are discouraged by many religions, similar messages could be tested for other faiths, where deemed appropriate, to determine if they help increase knowledge about the harms of smoking and encourage smoking cessation.

Saeed A.A., Khoja, T.A., Khan, S.B. Smoking behaviour and attitudes among adult Saudi nationals in Riyadh City, Saudi Arabia. Tob Control. 1996;5(3):215-9. El Awa, F. Middle East: religion against tobacco. Tob Control. 2003;12:245–250. Yong, H.H.; Fong, G.T.; Driezen, P.; Borland, R.; Quah, A.C.; Sirirassamee, B.; Hamann, S.; Omar, M. Adult Smokers’ Reactions to Pictorial Health Warning Labels on Cigarette Packs in Thailand and Moderating Effects of Type of Cigarette Smoked: Findings From the International Tobacco Control Southeast Asia Survey. Nicotine Tob Res. 2013;15(8):1339-47.

Additionally, more discussion would be appreciated regarding the choices of 3 specific messages used and the possible expansion of this work with additional health risks and differences between the specific messages (where one might be more impactful than another).

Response: Thank you for this feedback. We selected these three specific messages because suicide, harming yourself, and harming others are completely prohibited behaviors in Islam and smoking results in those outcomes. Other behaviors prohibited in Islam and linked to smoking could also be selected and used in the future studies.

We added the following paragraphs to the discussion and conclusion:

Discussion:

The central aims of the Islamic legal framework, as well as a majority of other religions, are to minimize the risk of harm to society and individuals and maximize the opportunities for collective and individual wellbeing. Considering this framework, we created three different text messages consisting of suicide, harming yourself, and harming others -- all of these are prohibited behaviors in Islam and could result from smoking. The created religious HWLs are designed to elicit feelings of guilt from self-harm (suicide and gangrene) and harming others (secondhand smoke).

Conclusion:

Considering other harmful health effects of smoking, this work could be expanded by additional religious textual messages stating other health risks of tobacco in combination with appropriate graphical images and tested with regard to their effectiveness.

Reviewer 2 Report

This is a well-written manuscript that examines the effectiveness of cigarette pack health warning labels with religious messages in an urban setting in Indonesia. This research presents the potential importance of religious-based tobacco control strategies. There are just a few limitations that, once addressed, will result in a manuscript that can greatly benefit public health.   

Comments:

While the cross-sectional design was mentioned in the title, it should be described in the method as well.

What is the Cronbach's alpha for the six HWL questions?

A table comparing the sociodemographic characteristics of the study participants by religiosity level (score 4 or 5 vs others, for example) should be given.

A stratified analysis of Table 3 by smoking status may help assess whether religious-based warning labels are more effective for smokers (to whom the labels are targeting) than non-smokers.

Limitations should address whether there were prior exposures of the same/similar labels by the participants before the study; and whether the study was able to differentiate the observed effect from the graphic pictures themselves (likely the dominant effect) and the addition of religious text.

Author Response

This is a well-written manuscript that examines the effectiveness of cigarette pack health warning labels with religious messages in an urban setting in Indonesia. This research presents the potential importance of religious-based tobacco control strategies. There are just a few limitations that, once addressed, will result in a manuscript that can greatly benefit public health.

While the cross-sectional design was mentioned in the title, it should be described in the method as well.

Response: We added “cross-sectional design” to the method section.

What is the Cronbach's alpha for the six HWL questions?

Response: We added the following statement to the method section:

The Cronbach's alpha for the six HWL questions was 0.961.”

A stratified analysis of Table 3 by smoking status may help assess whether religious-based warning labels are more effective for smokers (to whom the labels are targeting) than non-smokers.

Response: Thank you for this feedback. We had previously assessed whether sociodemographic characteristics were associated with rating the religious HWLs higher than non-religious HWLs. However, we did not find any significant differences, including for smoking status. Thus, we have not included a stratified analysis of Table 3 by smoking status.  However, if the editor would like us to include this table we would be happy to do it.

Limitations should address whether there were prior exposures of the same/similar labels by the participants before the study; and whether the study was able to differentiate the observed effect from the graphic pictures themselves (likely the dominant effect) and the addition of religious text.

Response: Indonesia implemented graphical health warning labels in 2014. In our study, all smokers stated prior exposure to similar images. Therefore, we do not think that prior exposure among smokers had a meaningful impact on our study findings.

We added the following statement to the limitation:

We did not test the effectiveness of textual message and graphical images separately; therefore, the current study was not able to differentiate the observed effect from the graphic pictures themselves and the addition of religious textual messages.”

A table comparing the sociodemographic characteristics of the study participants by religiosity level (score 4 or 5 vs others, for example) should be given.

Response: We added the following table and paragraph to the results.

“College graduates and occasionally/not at all smokers were more religious than primary/secondary graduates and daily smokers (p=0.049 and p=0.004, respectively). Participants who reported very good /excellent health status (88.9%) and between 3-5 million IDR monthly income (91.2%) had higher religiosity scores than participants who reported poor/fair health status (76.4%) and less than 3 million IDR monthly income (84.0%) (p=0.001 and p=0.040, respectively). There were no significant differences between religiosity level and gender, age, and having children (Table 2). In addition, we did not find any significant difference between rating score of religious and nonreligious HWLs and the sociodemographic characteristics of the participants.”

Table 2: Sociodemographic characteristics of participants by religiosity level

Religiosity Level

Characteristics

Score 1-3

Score 4-5

Gender

n

%

n

%

p

Male

84

13.0

560

87.0

0.590

Female

25

14.6

146

85.4

Age (Year)

18-24

65

15.5

354

84.5

0.065

≥25

44

11.1

352

88.9

Education Level

Primary/Secondary

16

21.1

60

78.9

0.049

High School

69

13.6

440

86.4

College

23

10.1

205

89.9

Having Children

No

74

13.8

462

86.2

0.606

Yes

34

12.5

238

87.5

Monthly Income (IDR)

<=3 million

67

16.0

352

84.0

0.040

>3 million and <=5 million

19

8.8

198

91.2

>5 million

17

13.3

111

86.7

Smoking Status

Daily

97

15.2

543

84.8

0.004

Occasionally/Not at all

12

6.9

163

93.1

Health Status

Poor/Fair

29

23.6

94

76.4

0.001

Good

48

11.2

380

88.8

Very Good/Excellent

26

11.1

208

88.9

Total

109

13.4

706

86.6

Reviewer 3 Report

Interesting paper that tries to measure personals believes (Muslim faith) and health behaviour (smoking).

The study is well designed and analysis is correct.

As a Public Health worker my major concern is related with the role of the religion in health care. In my opinion religion is a personal circumstance and it should be kept in the individual sphere and should not be used as a tool in the public sphere.

According to authors' conclusions religious and nonreligious HWLs were functionally equal in effectiveness. Why should religious HWLs be used then? Religious HWLs can disturb the health warnings for the whole population.

There are many risks using religion as a public health tool, as authors remarks at discussion chapter.

Author Response

Interesting paper that tries to measure personals believes (Muslim faith) and health behaviour (smoking).

The study is well designed and analysis is correct.

As a Public Health worker my major concern is related with the role of the religion in health care. In my opinion religion is a personal circumstance and it should be kept in the individual sphere and should not be used as a tool in the public sphere.

According to authors' conclusions religious and nonreligious HWLs were functionally equal in effectiveness. Why should religious HWLs be used then? Religious HWLs can disturb the health warnings for the whole population.

There are many risks using religion as a public health tool, as authors remarks at discussion chapter.

Response: Thank you for your feedback. It should be noted that it is not our intention to necessarily recommend religious-based labels at this time. We hold the same concern with regard to using religion as a public health tool. As noted by the reviewer, we included these concerns in the limitations. The potential benefits and harms would have to be evaluated and weighed carefully and comprehensively in each environment.

We added following statement to the limitation:

“This was an exploratory study tested under specific conditions for a specific group of people. Before a recommendation can be made about using religious messages on HWLs, it would be important to assess the magnitude of potential unintended consequences. For instance, there is a need to study and consider whether religious HWLs could have negative impacts on people of other faiths or people of no faith, particularly in states where power is concentrated amongst a single demographic or ruling class (e.g., persons of a single religion).”
